# Mentalization in Typically and Atypically Developing Iranian Children and Its Associations with Age, Sex, and Externalizing/Internalizing Symptoms

**DOI:** 10.3390/children10040657

**Published:** 2023-03-30

**Authors:** Masoumeh Zandpour, Jafar Hasani, Lyric Noelle Russo, Carla Sharp, Majse Lind, Jessica L. Borelli

**Affiliations:** 1Department of Psychology, Faculty of Humanities, Tarbiat Modares University, Tehran 14117-13116, Iran; 2Department of Clinical Psychology, Faculty of Psychology and Educational Sciences, Kharazmi University, Tehran 15719-14911, Iran; 3Department of Psychological Science, University of California, 4201 Social & Behavioral Sciences Gateway Irvine, Irvine, CA 92697, USA; 4Department of Psychology, University of Houston, 126 Heyne Building, Houston, TX 77024, USA; 5Department of Psychology, Aalborg University, 9000 Aalborg, Denmark

**Keywords:** mentalization, dimensions, internalizing symptom, externalizing symptom, typical, atypical developing children

## Abstract

Mentalization refers to the ability to understand the mental states of oneself and those of others that motivate action and behavior. Mentalization has generally been linked to adaptive development and healthy functioning whereas diminished mentalization has been associated with maladaptive development and psychopathology. The vast majority of research on mentalization and developmental trajectories, however, is based on Western countries. The overall aim of this study was therefore to examine mentalizing abilities in a novel sample of 153 typically developing and atypically developing Iranian children (Mage = 9.41, SDage = 1.10, Range = 8–11, 54.2% females) recruited from a primary school and health clinic in Tehran. The children completed semi-structured interviews that were later transcribed and coded for mentalization. The parents provided reports on internalizing and externalizing symptoms, demographic information, and all formal diagnoses of the children. The results pointed at general age and sex differences across the two groups. Older children showed more adaptive mentalization compared to the younger children; boys and girls used different mentalizing strategies when facing difficult situations. The typically developing children were better at mentalizing than the atypically developing children. Finally, more adaptive mentalization was associated with lower externalizing and internalizing symptoms among all children. The findings of this study contributes with expanding mentalization research to also encompass non-Western populations and the results hold crucial educational and therapeutic implications.

## 1. Introduction

Mentalization refers to one’s capacity to consider close relationships and the self in terms of mental states and see the actions and behaviors of others in terms of underlying psychological motivations [1]. Based on this model, from infancy onward, a child’s early sense of self develops around the experience of being treated as an autonomous being with a mind of their own by attachment figures and being responded to by caregivers as if their behavior and affect communicates something about their psychological experience [1,2]—by being treated as individuals with intent and agency, children are able to discover their minds and come to recognize themselves as having unique thoughts, feelings, and desires, all of which is necessary to reflect on their own mental processes and the mental processes of others [2,3].

Grounded in attachment theory, the capacity to mentalize, operationalized for research purposes as reflective functioning (hereby referred to as RF), is thought to develop through responsive caregiver-child interactions in which the parent focuses on the child’s emotions and motivations underlying the child’s behaviors. These responsive interactions enable the child to understand and, importantly, manage their own emotional and behavioral responses [3,4]. Further, as children age and are able to differentiate between the self and others, the capacity to mentalize grows [5]. High RF capacities are marked by efforts to tease out the mental states underlying behaviors, as well as by understanding the nature of mental states (e.g., their interdependence or ability to be disguised; [4]), factors that are integral to regulating one’s emotions and behaviors. Most critically, in reflective individuals, mental states are recognized as the key to understanding behavior in oneself or another, and as a result, internal working models of emotion and intentions emerge [4]. In contrast, RF impairment is characterized by a low understanding of internal states underlying behavior, as well as an unintegrated and base understanding of how emotions are dynamic aspects of experiences and relationships [5], and are thus associated with psychopathology and interpersonal and behavioral difficulties [6,7].

Theorists argue that the ability to reflect well is critical for resolving distressing experiences and promoting psychological well-being as it enables an individual to moderate the impact of life events and make sense of them [8] Although difficulties in mentalization are common during times of emotional overload or when the attachment system is triggered [9], consistent failures in mentalizing are thought to emerge in a broad range of psychopathologies in children [7,9,10]. Moreover, scholarship has begun to illuminate trends in child RF, particularly that adolescents with higher RF are more likely to be securely attached [11,12,13], have greater emotion regulation [14], have lower somatic symptoms [15] and experience less interpersonal difficulties [16,17]. Conversely, emerging findings have revealed that lower child RF confers risk for a variety of mental health outcomes in children and adolescents, including depression, externalizing, and somatic symptoms [18,19,20], although additional research is greatly needed.

Extant social neuroscientific literature has further indicated that the capacity to mentalize is underpinned by four dimensions that may directly influence psychopathology (See [2] for a review). These include: (1) Mentalizing with regard to the self and about others. Imbalance within this dimension occurs when an individual focus solely on the minds of others or, conversely, solely on their own needs and goals [21,22,23]; (2) Mentalizing based on external or internal features of the self and others. Self-other is a central dimension in RF as well as personality functioning. Psychological problems appear when an individual fails to balance internal and external awareness and therefore neglects the mental states of others [24,25]; (3) Cognitive versus affective mentalizing, which refers to the ability to name, identify, and reason about mental states (self or other), while imbalances in this dimension appear when the cognitive aspects are ignored or there is intense focus on emotions and cognitions [1,23]; and (4) Implicit (or automatic) versus explicit (or controlled) mentalizing. This type of mentalizing occurs when a person implicitly understands another person’s mind or uses his or her previous assumptions about others. It is reactive, instinctive and deliberately involves the rapid processing of social information. Conversely, explicit mentalizing refers to conscious action, such as when we talk about our feelings or motivations and control our body movements and the type of words we choose and the tone of our voice. Explicit mentalizing is a relatively gradual process that requires language, reflection, and mental effort. Mentalization problems occur when a person relies solely on automatic assumptions about their own or others’ mental states, which is very simplistic [22,24].

These dimensions are related and used simultaneously in the interpretation process [23] as they express the concept of how people perceive and interpret themselves and others in relation to themselves, as well as in interpersonal relationships and social situations [26]. Psychosocial functioning is thought to be achieved when individuals maintain a level of balance within these dimensions, as well as are able to shift their balance at each of the poles based on environmental needs and demands [23]. Conversely, if balance is not achieved, maladaptive self and interpersonal functioning ensues. In order for children to transcend a younger child personality and develop adult personality, there must be some degree of consistency between self-functioning and interpersonal functioning [27]. Recently, personality disorder has been reconceptualized to more adequately reflect a conception of maladaptive self and interpersonal functioning [28]. For example, the latest version of the ICD-11 [27] describes maladaptive self and interpersonal functioning as the entry criterion for personality disorder and the common feature shared across all manifestations of personality pathology, and may represent a general severity criterion that indicates high levels of psychopathology regardless of specific disorder [29]. As such, imbalanced mentalizing is thought to be associated with challenges in maladaptive self and interpersonal functioning that are specific to personality disorder, but may also be indicative of a wide variety of other psychopathologies [6]. At the same time, it is thought that different psychopathologies reflect different imbalances along these dimensions, resulting in various mentalizing profiles that are characteristic of specific disorders [2]. Research has documented associations between maladaptive mentalization and a wide range of disorders such as somatic disorder, destructive behaviors [20], borderline personality disorder (BPD), bulimia nervosa [30].

At present, few studies have shifted the lens to investigate the specific connections between child RF and mental health. In fact, research on child RF in children with mental health diagnoses has typically focused on general, self, or other RF [19,31,32,33,34], neglecting other proposed dimensions, despite the fact that theorizing regarding mentalizing is very rich, as described above. For example, in Bizzi and colleagues’ (2018) study of school-aged (8 to 15 year olds) children, they examined the links between self-focused and other-focused RF, finding that children with somatic symptom disorders and disruptive behavior disorders had significantly lower RF compared to children in the comparison group. Similarly, focusing on a sample of children exposed to childhood sexual abuse and controls, Ensink and colleagues (2016) [19] recruited 168 children aged 7 to 12 and their mothers, finding that self-focused and other focused mentalizing was inversely correlated with depressive symptoms and externalizing issues. These findings support the notion that mentalizing as a transdiagnostic risk factor that underlies many forms of mental health problems in children [34] but fail to elucidate the interrelation between the finer grained dimensions of RF and psychopathology. In sum, although previous studies have examined the relationship between RF and psychopathology, previous studies have examined children’s RF using this broader designation of self/other rather than a more fine-grained analysis. The use of a qualitative approach would enable a more nuanced evaluation of children’s RF in response to the CAI.

Further, additional research is needed to explicate the relationship between these dimensions within children who are underrepresented within the literature, as well as to examine how associations may differ as a result of child age, sex, mental health status, and culture. Although not yet studied, there are reasons to believe that mentalizing profiles may differ between cultures. For instance, individuals from individualistic (Western) cultures tend to focus on their own mental states more than the mental states of others. Whereas individuals from collectivist cultures (Eastern), such as Iran, in comparison, tend to focus more on others’ mental states compared to their own [35,36]. These differences may result in culturally-dependent mentalization profiles, such that children from individualistic cultures may demonstrate higher self RF and lower other RF, whereas children from collectivist cultures, may demonstrate lower self RF and higher other RF. Examination of such trends has the potential to provide insight into children’s mentalizing capacities that have wide clinical implications, as well as broaden the fields’ understanding of child mentalization across cultures. Research has shown that different cultures show different profiles of mentalization due to factors such as differences in cultural characteristics, linguistic factors, value preferences, and parenting characteristics [34]. This makes it necessary to adopt flexible approaches to measure RF in different cultures to reduce bias in studies, because existing categories encourage researchers to think and act within defined frameworks, but cannot assume that the coding categories would be similar in a different non-Western culture. Therefore, like other researchers who used qualitative approaches to analyze their findings, we relied on a qualitative approach [37,38].

Thus the current study aims to contribute to this gap in the literature by examining whether there are significant differences in mentalizing capacity and its four dimensions among typically developing (operationalized as not having a mental health diagnosis) and atypically developing (operationalized as having a mental health diagnosis) children in middle childhood (8–9 years of age) compared to late childhood (10–11 years of age), as well as whether differences appear as a result of child sex, in a sample of Iranian children. Today, extensive brain and genetic research has suggested that there are sex differences in humans. New findings in brain research have suggested that cross-cultural patterns of sex differences, genetic, hormonal and social influences in the expression of sex differences, as well as sex differences in the brain and cognition and human behavior and actions [39]. Iran is a patriarchal culture [40], so parents use different ways to raise girls and boys so that girls are more likely to be encouraged to be self-controlled and boys are encouraged to be assertive. These different styles of socialization can lead to changes in children’s social cognition. Children with different types of thinking patterns may be more or less likely to be interested in mental states, both as a function of their biological differences and their socialization history. As a result of the differences between socialization pattern in Iran and other countries, it cannot be assumed that mentalizing will work the same in Iran than in other countries.

To accomplish our study aims, Persian children completed semi-structured interviews that were later transcribed and coded to assess child RF and its dimensions. Parents, in turn, provided reports on their child’s internalizing and externalizing symptoms, demographic information, and all formal child diagnoses.

In addition, to ensure our sample of developing children did not meet diagnostic criteria, as well as to extend recent literature that has highlighted a significant association between child RF and internalizing and externalizing behaviors [18,19,20], we asked all parents to report on their child’s internalizing and externalizing behaviors. Associations of child RF and its dimensions in typically and atypically developing children were also evaluated. This information is pivotal as no study to our knowledge has examined child RF among non-Western populations, thereby obscuring the mentalizing experiences of children from Eastern cultures and how their mentalizing profiles may differentially develop in relation to psychopathology.

In sum, we sought to measure the dimensions of RF in a sample of Iranian children. We conducted our assessment using a qualitative approach without relying on existing schemes. We tested four hypotheses. First, we predicted that overall RF capacity and each of its dimensions would be greater among older children as compared to younger children (Hypothesis 1). Second, we expected that female children would be classified as having higher RF compared to male children on all dimensions of RF (Hypothesis 2). This hypothesis was based on extant social behavioral research indicating that across cultures girls are taught to prioritize emotional competency and socialization at a higher rate than boys [41]. Third, we predicted that typically developing children would be rated as having higher overall RF capacity (In this study, children’s RF is considered equal to children’s ability to reflect 4 dimensions of mentalization, and the dimensions of mentalization refer to each dimension independently of the other) compared to atypically developing children (Hypothesis 3). Finally, we predicted that parent-reported child internalizing and externalizing behaviors would be negatively correlated with child RF and its dimensions (for both typically and atypically developing children (Hypothesis 4), such that higher child RF across dimensions would be associated with less internalizing and externalizing behaviors. Understanding these connections, we argue, has the potential to inform future interventions and clinical work aimed at fostering psychological and relational health in both typically and atypically developing children.

## 2. Materials and Methods

### 2.1. Participants

The study was conducted in accordance with the Declaration of Helsinki, and the protocol was approved by the Institutional Ethical Committee at Kharazmi University IR.KHU.REC (2021, 1400.025). The study included 153 Persian children between the ages of 8 and 11 (Mage = 9.41, SDage = 1.10, 45.8% males, 54.2% females). Participants were recruited from a primary school (*n* = 122) and a community clinic (*n* = 31) to participate in a study on child development for children at risk for internalizing and externalizing disorders (e.g., anxiety, depression, behavioral problems). The participants were all residents of Tehran. In order to create maximum diversity in the participants, typically developing children were selected from several schools in Tehran (top schools with excellent educational quality, schools with average educational quality, and schools in the suburbs and poor areas with poor educational quality (as determined by Tehran’s Education Department). In total, children were recruited from 12 different schools in Tehran. To recruit typically developing children, after obtaining the necessary permits, a demographic questionnaire and the CBCL/6-18 ([42] 2001, Parent version) were sent electronically by the child’s school or community clinic to mothers who were potentially interested in having the child participate in the research, in Tehran. Parents interested in participating in the study provided contact information through an online questionnaire distributed by the child’s school or the community clinic, as well as provided information on their child’s current mental health status and diagnoses. Recruitment continued until data saturation was reached. See (Table 1) for more information about participant.

### 2.2. Procedure

After obtaining ethics approval through the Ethics Committee of Kharazmi University, as well as obtaining necessary permits, recruitment of participants and data collection began. The first author with participants who expressed interest in the study, contacted, then conducted initial screening to determine eligibility. The first author was introduced to the psychology clinic through the university. Also, through Education Department in Tehran, the first author was introduced to a number of elementary schools in different areas of the city. To recruit typically developing children, a demographic questionnaire and the CBCL/6-18 ([42], parent version) were sent electronically by the child’s school or community clinic to mothers potentially interested in having their children participate in the study. The purpose of CBCL was to screen children who, according to the mother, did not suffer from emotional-behavioral disorders, and their education and social performance was satisfactory. Due to the COVID-19 pandemic, parents were able to choose the CAI interview administration format, and the resulting sample included *n* = 95 interviews conducted over the phone, *n* = 54 in person, and *n* = 4 through video conference. The in-person CAI interview were completed in a private room at the community clinic or at the child’s school while their parents (predominantly mothers) were in a nearby room. All interviews were conducted in the official Farsi language. All parents had sent their written consent for their children’s participation in the research through an online questionnaire to the first author.

Prior to the commencement of data collection for the overarching study, a pilot study with eight interviews was conducted by the first author and mentored by a senior researcher. The purpose of this preliminary study was to become familiar with the interview and make any necessary adjustments before starting to collect large data. Answering CAI questions requires children to recall painful memories or talk about their privates, the children participating in this study sometimes showed emotional distress or resisted answering. Therefore, the children were indirectly asked questions with negative emotional charge. For example, instead of? Children were asked some children think that their parents do not love them. Have you ever felt this way? How do you think a child feels when he thinks his parents don’t love him?

In accordance with the Helsinki Ethics Statement, children could decide to withdraw from the study at any time and the interview was terminated if they did not wish to be interviewed. The interview lasted approximately 38 min to an hour. All interviews were conducted by the first author between February 2021 and October 2021. 

### 2.3. Measures

Children’s internalizing and externalizing problems were measured using the Child’s Behavior Checklist (CBCL/6-18; [42]). The checklist consists of 113 items and increases to 120 taking into account the open-ended questions. This checklist is typically completed by the parents or the person in charge of the child based on how the child has been feeling within the past 6 months. The CBCL is divided into two parts, with the first focusing on the child’s competences in various fields such as activities in school and social relations (e.g., disobedient at school) and the second focusing on emotional-behavioral problems (e.g., can’t get mind off thoughts). The internal consistency of this scale in the Iranian sample was good, α = 0.85 [43]. Cronbach’s alpha coefficient α = 0.67 for the internalizing scale and α = 0.89% for the externalizing scale.

Child internalizing problems. Parents reported on their child’s internalizing symptoms using the Persian version of the Child Behavior Checklist for ages 6–18 (CBCL/6–18; [43]), a 33-item assessment on whether their child displays a wide range of behaviors (e.g., from the internalizing broadband scale (e.g., sample item) in the past 6 months on a 3-point scale from 0 (Not true) to 2 (Very True or Often True). This measure has previously been validated and used in samples with Persian youth [41]. The Cronbach’s alpha in current study was good, α = 0.82.

Child externalizing problems. Parents reported on their child’s externalizing symptoms using the Persian version of the Child Behavior Checklist for ages 6–18 (CBCL/6–18; [41]), a 35-item assessment on whether their child displays any of a wide range of behaviors (e.g., from the externalizing problems broadband scale (e.g., argues a lot) in the past 6 months on a 3-point scale from 0 (Not true) to 2 (Very True or Often True). This measure has previously been validated and used in samples with Persian youth [43]. The Cronbach’s alpha in current study was good, α = 0.78.

Child Attachment Interview. Child reflective functioning was assessed using the Child Attachment Interview (CAI; [44]), and was used to code audio-recorded and transcribed data, CAI is a semi-structured interview for children aged 8 to 13 years. The CAI is a 15-item assessment that aims to activate the attachment system and elicit narratives about the child and their relationship with attachment figures. The CAI includes a set of questions about the child’s self-representation, representations of his/her primary caregivers, as well as questions about times of conflict, distress, illness, hurt, separation and loss). The predictive validity was between 69% to 72%, the internal consistency was 0.92, and the standard item alphas was moderate: in relation to mother 0.65 and father 0.55, and Inter-rater Reliability was 0.87, and stability coefficient were 0.63 [44]). For the purpose of the present study, the CAI was translated into Farsi and then back-translated to assure equivalence with the original version.

For the present study, CAI was the most suitable instrument that had the ability to measure RF and its dimensions [44]. Since there are no instructions for extracting dimensions of RF on the CAI, and considering the cultural differences we anticipated identifying in the dimensions of mentalization, we deemed it necessary to examine the dimensions of RF in Iranian children without imposing the frameworks obtained in the West. Therefore, we pursued a qualitative content analysis utilizing a deductive coding process to extract the RF dimensions from children’s narratives.

### 2.4. Coding

We used qualitative content analysis [45] to qualitatively analyze the content of the interviews with the CAI interviews to identify RF dimensions. This is a non-systematic method for deriving repeatable and valid inferences from the data. Its purpose is to achieve a condensed and broad description of the phenomenon and to reach the concepts or categories that describe the phenomenon [45]. Although there is no systematic rule for analysis in content analysis, the three-phases method is applied in order to extract categories from the data [45].

In step 1, involves the data collection, data familiarization, and selection of the units of analysis. In this first stage, the unit of analysis must be identified, so the coding team consisting of the original researcher, two doctors of clinical psychology who generally specialized in RF, and a methodologist chose the unit of analysis in a joint meeting. Since the unit of analysis could be a word or a subject [45], each dimension of mentalization was considered as a unit of analysis, so the word or sentences that reflected each dimension were chosen from verbatim.

In step 2, involves open coding, creating categories, and abstraction. In this phase, it is necessary for the researcher to get a general sense of the data by immersing himself/herself in the data [45]. So a researcher (in this case the first author of this paper) identified meaning units, then content-excludes them without losing the original meaning and content, and regularly compared and reviewed them with the original text. Then, in consultation with the coding team, the researcher prepared the codebook. In order to increase the reliability of the coding, initially 30 interviews (original interviews, and content-exclude meaning units, and codebook), i.e., about 20% of the total interviews, were sent to two professors of clinical psychology at Tarbiat Modares University who were experts in mentalization. The coding team read the verbatim several times, then made open coding the data, categorizing the data belonging to each RF dimension into smaller categories, and finally formulating a general description of the research topic through categorization the initial code (abstraction). The coding process was started simultaneously by the first author and two other raters, then the resulting codes were reviewed by the first author and the interclass correlation coefficient, ICC, was calculated. We calculated interrater reliability on the ratings for this subset of the verbatim that had been coded, which were good; ICC for single measures = 0.77 and for Average measures = 0.91. Therefore, due to the acceptable reliability of the coding, the rest of the interviews were coded by the initial researcher.

Step 3, in which the researchers produce a valid and concise report of the analysis process and the steps taken to extract the dimensions of mentalization was prepared.

It should be noted that for the reliability of the analysis and to ensure the accuracy of the stages performed, in each stage a complete report of the process and how to perform the analysis was sent by the initial researcher to the other two raters. It is worth noting that each of the dimensions of mentalization encompasses a wide range of concepts, therefore, in this study, only parts of each dimension have been studied. To identify the topics covered in this study, see Codebook in Table 2.

The unit of analysis was the child’s response to each question, and the codes of each interview were extracted separately. Then, based on how many times in each interview the child reflected on each of the 8 dimensions of mentalization, the frequency of each dimension in each interview was counted and entered in SPSS software version 25 for quantitative analysis.

## 3. Data Analytic Procedure

Analyses were conducted using IBM SPSS Statistics 23.0. There was no missing data on the questionnaire packet. First, to test hypothesis 1–3, three multivariate analysis of covariance (MANCOVA) were executed with the age groups (8 to 9 years old versus 10 to 11 years old), sex and psychopathology as the independent variables, and the means of the eight categories of the RF as dependent variables. Before conducting MANOVA, the typical assumptions of an MANOVA including normality, equality of variance, and univariate outliers and additional assumptions such as absence of multivariate outliers, linearity absence of multicollinearity and equality of covariance matrices checked and obtained.

Second, to test Hypothesis 4, stepwise multivariate regression analysis (MRA) was used to test the association between RF Dimensions as independent variables and outcome variables as dependent variables (internalizing and externalizing psychopathology symptoms).

According to [46], the assumptions of multiple regression, including linearity, independence, homoscedasticity, normality, and multicollinearity were obtained. Overall, a visual inspection of the scatterplot indicated that the assumption of linearity was met for all dependent variables. Normality was tested by visual inspection of a histogram and analysis of the Shapiro-Wilk Test for each criterion variable. Moreover, visual inspection of the scatterplot of residuals versus predicted values and the Durbin-Watson Test (less than 4) for all criterion variables were analyzed to check for homoscedasticity. While some of the data points appeared to have more variability than others, there was no overall pattern. In addition, the majority of the residuals fell between 2 and −2. Generally speaking, these results suggest that homoscedasticity was not violated for either criterion variables, as the variability of the residuals must be robust to violate this assumption [42]. Finally, multicollinearity was tested by examining the Variance Inflation Factor (VIF) and tolerance values. Lack of multicollinearity was met as the VIF values were less than 10 and tolerance coefficients were less than 0.10.

## 4. Results

### 4.1. Hypothesis 1: Older Children Will Report Higher RF Capacity Compared to Younger Children

As mentioned above, we predicted that that children in late childhood would report higher RF capacity compared to children in middle childhood, regardless of whether their parents indicated they had a mental health diagnosis (Hypothesis 1). Results of a MANOVA displayed that there was a significant overall difference (Wilks λ = 0.62; F (8, 113) = 8.53, *p* = 0.000, η2 = 0.38) in RF dimensions between two age groups (8 to 9 years old versus 10 to 11 years old). Comparing the estimated marginal means (Table 3) indicated that the 10 to 11 years old group demonstrated significantly higher implicit, external, affective and cognitive dimensions of RF than the 8 to 9 years old group. Moreover, differences between two groups were non-significant for self, other, explicit and internal dimensions of RF.

### 4.2. Hypothesis 2: Girls Will Show Higher Levels of RF Capacity Compared to Boys across Dimensions

To assess Hypothesis 2, we tested sex differences in all dimensions of RF, mainly in typically developing children. Results of a MANOVA displayed that there was a significant overall difference (Wilks λ = 0.86; F (8, 113) = 2.31, *p* = 0.025, η2 = 0.14) in RF dimensions between two sex groups. Comparing the estimated marginal means (Table 4) indicated that the girls demonstrated significantly higher explicit dimensions of RF than boys did whereas the boys reported using significantly more affective dimensions of RF than girls did. Moreover, differences between two groups were non-significant for self, other, implicit, internal, external and cognitive dimensions of RF.

### 4.3. Hypothesis 3: Typically Developing Children Would Report Higher RF Capacity Compared Atypical Developing Children

To assess hypothesis 3, we tested all dimensions of RF in typically developing children and atypically developing children. Results of a MANCOVA exhibited that there was a significant overall difference (Wilks λ =0.74; F (8, 144) = 6.23, *p* = 0.000, η2 = 0.26) in RF dimensions between typically developing children and atypically developing children. A comparison of the estimated marginal means (Table 5) indicated that the typically developing children group demonstrated significantly higher self, implicit, internal, affective and cognitive dimensions of RF than atypically developing children group. Moreover, differences between two groups were non-significant for other, explicit and internal dimensions of RF.

### 4.4. Hypothesis 4: Child Internalizing and Externalizing Behaviors Based on CBCL Scores Would Be Negatively Correlated with Child RF and Its Dimensions for Both Typically and Atypically Developing Children

We examined the bivariate associations between the RF dimensions and outcome measures (internalized and externalized psychopathology symptoms), which enabled us to assess the individual associations between these variables. In addition, in order to examine whether each individual’s RF dimensions was significantly associated with outcome measures after controlling for the other scale, stepwise regressions were conducted while controlling for age and sex. As shown in Table 5, self and cognitive dimensions of RFC were negatively associated with internalizing symptoms. Moreover, self, other and cognitive dimensions of RFC were negatively associated with externalizing symptoms. In addition, the results presented in Table 6 show that the associations between outcome measures with RCF scales—even after controlling age and sex—remained significant.

In addition, the bivariate associations between the RF dimensions and outcome measures (internalized and externalized psychopathology symptoms) were investigated according to separate age groups (8, 9, 10, 11 years old). These results are presented in Table 7.

## 5. Discussion

This is the first study to examine dimensions of mentalization in a sample of typically developing and atypically developing Iranian children aged 8 to 11 years. We examined the age and sex differences of children in mentalization and its dimensions, and also compared the age and sex differences of typically and atypically developing children on RF dimensions. The results showed that in the two groups of typically and atypically developing children, there were significant age and sex differences in RF and its dimensions and greater mentalization was associated with lower externalizing and internalizing symptoms. The results indicated that mentalization as a concept is particularly relevant and important to assess in typically and atypically developing children to capture variations in mentalizing profiles among children alongside the cultural component.

Consistent with the research literature, our first hypothesis was confirmed. Findings from the age differences of the children indicate that children aged 10 to 11 responded more reflectively to CAI questions than children aged 8 and 9 years. This finding indicates that higher mentalizing capacity in older children means that older children are better at making sense of their internal subjective experience and can use verbal language effectively to express this. Younger children have less meta-cognitive capacity and more limited verbal ability to express and communicate complex feelings about themselves and their families [6]. Children’s differences aged 10 to 11 years compared to children aged 8 to 9 in the external dimension may indicate that older children are more able to consider and respond to the internal needs and desires of others [6]. Also, older children are more capable to recognize inner states and other emotions and feelings from face and body postures. Research has shown that emotions are visible and transferable from the face and body postures, and children have different abilities to recognize these emotions based on age, and with age, the ability to recognize all emotions in children increases [47]. Children’s ability to recognize emotions from facial and body expressions has a positive effect on children’s social competence and cognitive development [48,49,50], because it enables children to use this ability to create related attributions [51], which affects their social performance and interpersonal relationships. These findings suggest that older children have a greater ability to balance the affective-cognitive pole compared to younger children and they are able to move and maintain balance with greater flexibility in these dimensions. These findings indicate that with age, the ability to recognize distinct emotions [44], as well as RF capacity in children may increase.

Our second hypothesis posited that girls would show superior RF capacity relative to boys. In support of this hypothesis, our findings revealed that girls have higher capacity for explicit mentalizing than boys and do more deliberate processing in the face of difficult situations than boys. Instead of responding quickly and implicitly to situations, girls engage in more mental processing than boys. This finding may be largely related to differences in sex abilities in language skills and vocabulary. Studies have evidenced a relationship between language and success in theory of mind tasks [52]. Although we did not formally assess language or vocabulary skills using standardized tests, our informal observations using the qualitative analysis were that the girls in this study showed more verbal and vocabulary skills than the boys. They were able to provide more explanations and also, girls showed more interest and comfort in the interviews compared to boys. In contrast, our qualitative analyses showed that boys provided more emotional responses to difficult situations than girls, which means that boys’ responses to difficult situations were more impulsive and they expressed more affect than girls and with less cognitive processing of the situation. There seem to be sex differences in the response to maltreatment (items such as, have you ever been hit by a grown up in your family?) between girls and boys, and the two sexes use different strategies to overcome stress [53]. Biologically-based temperamental predispositions and different socialization of girls and boys effect how emotions are expressed [54]. Girls’ potential biological abilities in language and self-regulation abilities lead girls to reduce anger and other external emotions and increase positive emotional expression [50], which may make girls more inclined to respond according to sex roles and adherence to methods that maintain relationships [54]. Girls are also more able to hide negative emotions, while boys are allowed to express external feelings such as anger and disgust more than girls and show less tender feelings than girls. Therefore, they show more negative emotions and less verbal ability and inhibitory control than girls [54]. It is worth noting that many aspects of cognition and social cognition show different age and cultural interactions, and this complicates the study of these phenomena [55].

Our third hypothesis involved predicting differences in RF between typically and atypically developing children. The analyses showed that typically developing children were significantly more able to reflect in terms of the internal, implicit, affective and cognitive dimensions compared to atypically developing children [2]. Since deficits in mentalization are associated with psychopathology [26] lower levels of mentalizing capacities may be a cause for concern. At the current time, we do not have norms for mentalizing within this age range, so we do not know what constitutes a low level of mentalizing, but future developmental work on this topic will be important in order to establish benchmarks in this regard.

In the fourth hypothesis, we examined the relationship between the RF and internalizing and externalizing disorders in typically and atypically developing children. The results showed that self and cognitive dimensions have a negative association with internalizing disorders. Relatedly, Ref. [56] found a positive association between externalizing and internalizing disorders with mentalization problems. They showed that children with internalizing problems undermentalize self and their emotions, but children with externalizing problems suffered from severe deficits in mentalization. Internalizing symptoms are also related to social-cognitive biases in relation to the self and attempts to withdraw from physical sensations and other internal experiences (higher levels of experiential avoidance) [57].

Children’s defects in mentalization and its dimensions in atypically developing children, on the one hand, create significant difficulties to create rich and understanding relationships with others. On the other hand, it hinders the acquisition of a coherent and integrated sense of self in these children. This reveals the importance of adaptive self function related to interpersonal adaptive function for personality development and transformation from child to adult personality function. Disruption of these functions increases children’s vulnerability to personality functioning and personality disorder, especially BPD [27], because the organization of behavior is explained based on a two-factor model of internalizing and externalizing behaviors [27]. Some researchers argue that a central function of internalizing and externalizing behaviors is to express feelings in the context of relationships [28]. Although internalizing and externalizing symptoms are not themselves associated with personality pathology in childhood, they confer risk for personality pathology later in development, such as in adolescence [27,28]. When children enter adolescence and the need to acquire cultural skills and competences to successfully transfer to the world of adulthood and take on the role of an adult, the capacity for personality dysfunction appears [28].

This is the first study to investigate RF and its underlying dimensions among a sample of Iranian children. Also, in previous studies, existing categories were used to measure RF, but we here we endeavored identify RF dimensions in Iranian children using a qualitative approach, without imposing Western schemes. More studies are needed to determine the relationship and impact of variability in mentalizing dimensions and its associations with developmental psychopathology.

### 5.1. Research Limitation

Research on mentalization has only recently gained ground in Middle Eastern countries. The novelty of this sample and the limited extant literature therefore gave rise to some challenges when designing the study and generating hypotheses. For instance, we did not evaluate the effects of interview delivery format (e.g., phone, in person) on the quality and quantity of children’s responses. Another limitation was the somewhat small sample size, which potentially reduced our power to detect significant effects. The another limitation was related to the fact that mostly mothers cooperated with the research, not both parents. This is mainly due to the fact that in Iran mothers are generally more involved in their children’s daily life and well-being and were therefore more likely to take part of the research study. However, this limits the information we received about the children. Another limitation was that we did not use a tool for our RF validation, so we do not have data that can be used to validate our RF coding scheme. An additional limitation was the absence of diagnostic interviews to determine the absence of psychopathology in the typically developing children. The final limitation of the study was that due to the exploratory nature of our study we did not use post hoc tests to correct false positives. Future work is needed to elaborate on temporal dynamics between mentalization, age, sex and externalizing/internalizing symptoms.

### 5.2. Concluding Remarks

Research in mentalization refers to studying the capacity to understand the mind of the self and those of others and has only recently gained ground in Middle Eastern countries. This study investigated 8–11-year-old children’s mentalization and its dimensions in typically and atypically developing children from Iran. Our findings were largely consistent with our predictions and with theorizing. We identified age and sex difference in RF, and also learned that typically developing children had more advanced RF than atypically developing children. Finally, higher RF was associated with lower externalizing and internalizing symptoms.

The study is an important step towards broadening our knowledge of mentalization to also encompass non-Westernized countries. The findings of this study can have fundamental implications for child psychotherapists. First, these results will provide some insights to Mentalization-Based Treatment for Children (MBT-C), suggesting that child age and sex be taken into account when developing treatment protocols. Second, the links between clinical group and psychopathology with RF underscore the need for mentalization-based intervention programs for children in Iran.

## Figures and Tables

**Table 1 children-10-00657-t001:** Demographic Information of Participants.

N = 153	Typically Developing = 122		Atypically Developing = 31	
Sex Frequency in each group	Male = 54	Female = 68	Male = 16	Female = 15
Age Frequency in each group	8 = 15	8 = 19	8 = 4	8 = 4
	9 = 13	9 = 15	9 = 5	9 = 3
	10 = 15	10 = 19	10 = 3	10 = 3
	11 = 11	11 = 15	11 = 4	11 = 5

**Table 2 children-10-00657-t002:** Codebook used to extract the dimensions of mentalization from children’s narratives.

Self: The child is self-oriented. The child talks more about him/herself, looks more at the situation from his/her own perspective, and considers it. Although he/she may shift to other dimensions, they return to him/herself. The child is very self-focused.
Other: The child is other-oriented. He/she talks more about the other, looks at the situation from another perspective, and considers it. Although he/she may shift to other dimension, he/she returns to other. The child is very focused on the other.
Internal: (a) The child considers most of his/her inner world (desires, feelings, needs and beliefs) (b), in the face of a difficult situation, uses less external signs of communication to enhance and correct his/her perception of the situation, and uses more of his/her inner states, such as his/her intuition, to understand the situation.
External: (a) The child considers most of the other inner world (desires, feelings, needs and beliefs) (b), in terms of the situation, uses external signs of communication (such as facial expressions, and body language) to enhance and correct his/her perception of the situation.
Implicit: child responses are fast, uncontrolled, and automatic. The child seems to be consciously or unconsciously trying to hide his/her mental content. The child tries to avoid answering by talking about irrelevant things and speaking about tangential topics.
Explicit: The child’s answers are accompanied by reflection and thinking effort. The child tries to remember the situation and thinks about it. The child’s narratives are accompanied by relevant explanations of the question asked.
Affective: The child’s responses are impulsive. The child responds based on the feelings and emotions he/she experiences in the moment.
Cognitive: The child’s answers are thoughtful. The child considers different aspects of the situation and tries to give more logical answers by controlling his/her emotions.

**Table 3 children-10-00657-t003:** Means, Standard Deviations, and Age Group Differences in CRF dimensions.

	8 to 9 Years Old (*n* = 61)		10 to 11 Years Old (*n* = 61)		F	*p*	η2
	M	SE	M	SE			
Self	13.74	2.84	13.80	2.84	0.01	0.90	0.000
Other	1.97	1.68	1.98	1.96	0.002	0.96	0.000
Implicit	2.24	1.99	3.21	2.61	5.29	0.02	0.04
Explicit	14.33	3.36	14.84	3.22	0.72	0.40	0.006
Internal	8.88	3.28	9.23	3.34	0.32	0.57	0.003
External	0.47	0.81	1.55	2.09	14.17	0.000	0.11
Affective	10.67	3.13	12.57	2.98	11.79	0.001	0.09
Cognitive	1.51	1.97	3.55	2.71	22.79	0.000	0.16

**Table 4 children-10-00657-t004:** Means, Standard Deviations, and sex Differences in CRF dimensions.

	Boy (*n* = 54)		Girl (*n* = 68)		F	*p*	η2
	M	SE	M	SE			
Self	13.44	3.13	14.02	2.77	1.19	0.28	0.01
Other	2.25	2.05	1.75	1.58	2.38	0.12	0.01
Implicit	3.01	2.52	2.50	2.21	1.45	0.23	0.01
Explicit	13.65	3.71	15.23	2.72	8.26	0.005	0.06
Internal	9.14	3.26	8.98	3.39	0.07	0.79	0.001
External	1.13	1.78	0.92	1.58	0.44	0.51	0.004
Affective	12.39	3.12	11.01	3.14	5.80	0.02	0.05
Cognitive	2.92	3.25	2.22	1.83	2.28	0.13	0.02

**Table 5 children-10-00657-t005:** Means, Standard Deviations, and sex Differences in CRF dimensions.

	Typically Developing Children (*n* = 54)		Atypically Developing Children (*n* = 68)		F	*p*	η2
	M	SE	M	SE			
Self	13.77	2.94	11.71	2.34	13.25	0.000	0.08
Other	1.97	1.82	1.97	1.49	0.000	0.98	0.000
Implicit	2.73	2.36	0.97	0.91	16.52	0.000	0.10
Explicit	14.58	3.29	13.81	3.41	1.35	0.25	0.009
Internal	9.05	3.33	6.74	2.82	12.69	0.000	0.08
External	1.02	1.67	0.71	1.75	0.82	0.37	0.005
Affective	11.62	3.19	9.54	2.53	11.28	0.001	0.07
Cognitive	2.53	3.57	0.90	1.16	11.75	0.001	0.07

**Table 6 children-10-00657-t006:** Pearson correlation between CRF dimensions with Internalized and externalized psychopathology symptoms and stepwise MRA results after controlling age and sex.

CRF Subscale	Internalized Symptoms					Externalized Symptoms				
	Pearson r	MRA				Pearson	MRA			
	r	B	R^2^	SE	95% CI	r	B	R^2^	SE	95% CI
Self	−0.23 ***	−0.16 *	0.19 ***	0.16	−0.67, −0.03	−0.08				
Other	−0.09					−0.30 ***	−0.20 ***	0.19 ***	0.38	−1.78, −0.28
Implicit	−0.03					0.09				
Explicit	−0.04					−0.05				
Internal	−0.16					−0.11				
External	−0.13					−0.30 ***	−0.17 **	0.22 ***	0.40	−1.71, −0.13
Affective	−0.05					−0.09				
Cognitive	−0.40 ***	−0.37 ***	0.16 ***	0.19	−1.36, −0.58	−0.36 ***	−0.33	0.13 ***	0.27	−1.75, −0.68

Note: * = 0.05; ** = *p* < 0.01; *** = *p* < 0.001.

**Table 7 children-10-00657-t007:** Pearson correlation between CRF dimensions with Internalized and externalized psychopathology symptoms in four age groups.

	8 Years Old (*n* = 42)		9 Years Old (*n* = 39)		10 Years Old (*n* = 40)		11 Years Old (*n* = 32)	
	Internalized Symptoms	Externalized Symptoms	Internalized Symptoms	Externalized Symptoms	Internalized Symptoms	Externalized Symptoms	Internalized Symptoms	Externalized Symptoms
Self	−0.344 *	−0.150	0.114	−0.078	−0.065	−0.177	−0.044	0.039
Other	0.028	−0.079	−0.157	−0.165	−0.290	0.019	−0.087	−0.254
Implicit	0.141	0.133	0.142	0.318 *	−0.171	0.003	0.371 *	0.425 **
Explicit	−0.220	−0.211	0.267	−0.055	−0.036	−0.398 *	−0.037	0.054
Internal	−0.485 ***	−0.224	0.047	−0.085	−0.064	−0.165	−0.109	0.040
External	0.166	0.148	0.008	−0.002	0.010	0.053	−0.055	−0.204
Affective	−0.008	0.159	−0.030	−0.005	0.122	0.211	0.121	0.262
Cognitive	−0.095	−0.249	−0.021	−0.035	−0.014	0.222	−0.303	−0.309

Note: * = 0.05; ** = *p* < 0.01; *** = *p* < 0.001.

## Data Availability

The data are available upon request to the first author.

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
