# Peer review of "Mentalization in Typically and Atypically Developing Iranian Children and Its Associations with Age, Sex, and Externalizing/Internalizing Symptoms"

_children, 2023, doi:10.3390/children10040657_

Round 1

Reviewer 1 Report

The study aimed at investigating mentalization in two groups of Iranian children with typical development or atypical development. The research involved a sample of 153 children (aging between 8 and 11 years): They were administered a semi-structured interview, namely the Child Attachment Interview (CAI). Parents provided information about demographics, formal diagnosis, and internalizing versus externalizing symptoms (this latter by means of the Child Behavior Checklist for ages 6-18 (CBCL/6-18). Results showed age, gender, and group differences.

The topic of the paper is quite interesting. However, there are several points I would like to see amended in a careful revision of the manuscript. First of all, I believe that a thorough language review could help the reader in fully understanding the text.

Introduction

In my opinion, the Introduction should be shortened, avoiding unnecessary repetition. As the authors focus on the role of mentalization in maladaptive behaviors among children, they should clarify which disorders depend on (or are associated with) poor mentalization. As the authors focus on the role of mentalization in maladaptive behaviors among children, they should clarify which disorders depend on (or are associated with) poor mentalization. More information should be provided about the studies by Bizzi et al. (2018) and Ensink et al. (2016), such as, age range of participants, type of psychopathology they suffered from, instruments used to measure mentalization, among others That mentalization varies according to developmental trajectory, gender, and cultural context is quite obvious or, at any rate, already demonstrated in the extant literature. Although I agree with what the authors say about the role of culture in mentalization, I think it may be misleading to implicitly suggest that all the tools used so far were quantitative in nature. The Authors stated: “Thus the current study aims to contribute to this gap in the literature by examining whether there are significant differences in mentalizing capacity and its four dimensions among typically developing (operationalized as not having a mental health diagnosis) and atypically developing (operationalized as having a mental health diagnosis) children in middle childhood (8-9 years of age) compared to late childhood (10-11 years of age), as  well as whether differences appear as a result of child sex, in a sample of Iranian children” (p. 4 of the manuscript). Given that Bizzi et al. (2016) compared a clinical group with a control group, the Authors should specify how their study differs from other papers already published. In addition, before defining the aim of the paper, it would be useful to dwell on the role of gender in mentalization processes. By the way, the Authors used the terms "sex" and "gender" indifferently. Is there a specific reason for this choice? I have no doubt to believe that this is the first study to analyse the phenomenon of mentalization among Iranian children. However, since it is not a cross-cultural investigation, it cannot provide information about the actual differences in mentalization due to cultural factors. Several studies have provided strong empirical evidence demonstrating that mentalizing abilities improve with age. So the first hypothesis sound somewhat obvious. The same applies to the third hypothesis. The authors did not analyse the role of different pathologies on mentalization processes, but rather focus on comparing children with typical development or atypical development. To expect differences between the two groups regardless of the psychopathology of the clinical sample risks confusing cause with effect.

Materials and method

Participants

I could not figure out how many subjects in the control and clinical group belonged to the two different age groups. It is unclear how the absence of psychopathological traits was detected in the control group. In other words, how reliable were the parents' statements considered to be?

Procedure

Did the authors evaluate the effects of the format of interview administration (by phone, in person, by video conference)? Were both parents interviewed for each participant or only one? If yes, the father or mother predominantly? This aspect is not clear to me.

Measures

Coding

The Authors devoted much space to illustrating coding procedures, but they did not describe the CAI as accurately. It would be erroneous to assume that the reader knows the tool precisely. The same holds true for the CBCL/6-18.

Data analytic procedure

Regarding age groups, the authors used the 8-9 and 10-11 age groups. In my opinion, it would have been better to use age as a continuous variable (8, 9,10, and 11 years) in their statistical analyses. In addition, information about the school attended was not reported. This variable is not necessarily reflected by chronological age. Next, regarding the regression analysis, the authors should specify the dependent variable and independent variables included in the analysis and the steps of the analysis.

Results

Considering the weakness of the hypotheses, the reader is unable to grasp the novelty of the findings.

Discussion

In my opinion, the discussion is somewhat repetitive. Mostly it did not emphasize the novelty (if any) of this contribution.

References

I noted an imperfect correspondence between text and references.

Author Response

Dear Reviewer, We the authors sincerely appreciate you for giving us the opportunity to receive your very helpful comments. Thank you!

Please See the attachment.

Reviewer 2 Report

Thank you for the opportunity to review the manuscript.

The work submitted for review examines a topic of great relevance in the field of psychology. The topic is very important, and the conceptual analysis made in the text is quite deep. The literature consulted is quite current but the sample isn´t quite large (which is a limitation for your work.). It should be analyzed with the chi-square statistic if there are no statistically significant differences in the sample due to gender issues. In such a case, the statements made in the conclusions are not entirely true. I would like to thank the efforts by the authors of the manuscript and congratulate them on the work. Overall, the writing is clear, the goals are well described, the introduction should explain the objectives of the study based on the review of the previous literature and the conclusions are properly made and presented. I consider that the constructs proposed in the abstract of the work are quite well explained. Therefore, the manuscript brings significant knowledge of the scientific literature so and still covers existing gaps in the field. On a formal level, the manuscript complies with the requirements of the Journal and references are written in accordance with the regulations of the Journal. The work is ambitious, and the results confirm most of the hypotheses and the relevance and potential of the work is therefore recognized, but this Reviewer considers that several changes are needed to the manuscript is publishable. In this sense, it should better explain the novelty and relevance of the work considering the previous empirical evidence and should better describe the practical implications. The process for selecting participants and the procedure should be better described. The study hypotheses should also be better explained. It should better describe the measuring instruments, providing some examples of items for each one. On the other hand, when they talk about gender differences, they are really sex differences (men vs women). It should describe the discussion and conclusions of the work better and, above all, update the manuscript references (most should be from the last 5 years). Finally, I wish the Authors the best in continuing this line of research.

Best wishes for Authors.

Author Response

Dear Reviewer, We authors sincerely appreciate you for giving us the opportunity to receive  your really helpful comments. Thank you!

Reviewer 3 Report

This paper explores mentalization in Persian children with different developmental trajectories. The work is properly done, and the results significantly contribute to the literature. The associations among mentalization and externalizing/internalizing behavior are consistent with the existing literature. 

Comments and questions: 

- Why did the authors use only a single measure to assess mentalization? This is a limitation of the study. 

- Are there any data on the association between mentalization from the attachment interview and other sources (e.g., specific questionnaires or interviews designed to focus on mentalization)? 

- Did the authors use corrections for multiple comparisons for the tests shown in Table 2-4?

- What are the most important take home messages from this study for the clinician? 

Author Response

Dear Reviewer, We the authors sincerely appreciate you for giving us the opportunity to receive your really helpful comments. Thank you!

Round 2

Reviewer 1 Report

The authors did their best to improve the paper. The manuscript is now suitable for publication

Reviewer 2 Report

Thank you for the opportunity to review the manuscript again.

Overall, the writing is clear, the goals are well described, well-considered introduction and the results properly made and presented. Therefore, the manuscript brings significant knowledge of the scientific literature so and still covers existing gaps in the field of Education. Therefore, my assessment is positive for the publication of this work, with a new suggestion. I would like to thank the efforts by the authors of the manuscript and congratulate them on the work. I recognize that they have considered almost all considerations of the Reviewers. Clearly, all the comments from Reviewers have contributed to a better quality of the manuscript. I have checked in the revised manuscript are corrected the most of errors found by the reviewers, both formally and content.

My best wishes to the authors.